# The Effectiveness of Applying Artificial Intelligence in Sick Children’s Communication

**DOI:** 10.3390/bioengineering11111097

**Published:** 2024-10-31

**Authors:** Hsin-Shu Huang, Bih-O Lee

**Affiliations:** 1Department of Nursing, Central Taiwan University of Science and Technology, Taichung 40601, Taiwan; 2College of Nursing, Kaohsiung Medical University, Kaohsiung 80708, Taiwan; biholee@kmu.edu.tw

**Keywords:** artificial intelligence, interactive learning, visual teaching materials, personalized e-learning materials

## Abstract

Pediatric nursing students are required to be taught how to overcome their own psychological stress during their internship in order to understand sick children’s emotional reactions, as well as to be able to interact and communicate with such children. This quasi-experimental study proves that the application of AI image health education e-books by the nursing teacher is more effective than traditional paper handout teaching materials in improving nursing students’ self-efficacy when using therapeutic games to deal with and reduce sick children’s fears of medical examinations and treatments (*p* < 0.05). AI-driven tools can enable the development of personalized e-learning materials that target specific areas for cognitive improvement. This targeted approach can enhance knowledge retention and skill development, resulting in better-prepared healthcare professionals.

## 1. Introduction

The majority of pediatric care patients are in their early childhood and preschool years. They are more prone to exhibiting fearful behaviors, such as crying and rejection, before medical examinations and treatments as they are too young, have poor oral expression skills, have had bad medical experiences in the past, are being exposed to unfamiliar medical staff, and/or have a lack of unself-conscious personal control, which forms the biggest source of psychological stress for nursing students during pediatric internships. If sick children’s fears are not handled in time, they will be prone to low self-esteem, insecurity, and lacking self-confidence, and this will incur negative impacts on their mental health and personality formation [1]. Psychologists Evie Crotty and Alberto Magni pointed out that children’s fearful emotions should be understood to help them face and overcome their fears [2]. Therefore, when sick children receive medical treatment, the timely improvement of their behavioral responses to fear is the goal of pediatric care. Therapeutic games are an important and effective intervention measure that helps medical staff to observe sick children’s physical and mental needs and feelings, help to mitigate their fear and stress, and improve sick children’s ability to obtain a sense of control over the treatment process [3].

Most nursing students express that, before their clinical internship in pediatrics, they lacked motivation to learn about child development, had never learned it before, had a vague sense thereof, and/or could not apply it to clinical communication. As a result, they had a poor response ability to cope with preschool children’s fear of medical examinations and treatments, and did not even have confidence in themselves. The higher their learning motivation, the more likely they are to use learning strategies, and the higher their learning efficiency. Thus, the clinical instructor plays a key role in stimulating learning motivation and formulating learning strategies simultaneously. The nursing teacher and students applied AI image health education e-books of therapeutic games that are suitable for pediatric clinical nursing based on the literature, pediatric clinical practice, and the learning needs of nursing students. Therapeutic games could be applied in nursing interventions from the beginning of internships for the purpose of helping nursing students, families, and children to communicate smoothly, as well as in developing friendly relationships. It was expected that the nursing students would build confidence in the pediatric internship process and would be willing to continue to use it in future clinical scenarios.

Pediatric patients have higher levels of emotional responses when facing unfamiliar environments and medical staff than adults [4], and this is a common challenge for nursing students on internships in pediatric clinical practice. People of different ages have different desires and needs with respect to using different communication methods, or they may need assistance in handling other family-member problems; having said this, however, nursing students on internships in pediatric clinical practice also look forward to participating in clinical nursing jobs. Nursing students hope that they will be able to communicate smoothly with children and that they can face challenges from family members professionally, as well as earning recognition and trust from their instructors [5].

Going through the internship experience is the best way through which to develop nursing professionalism [6]. Clinical instructors serve as professional models, assisting the students on internships in developing their skills and providing professional guidance (where academic theory and clinical practice are combined simultaneously, thus becoming the factor that has the most impact on students [6,7]), as well as providing a certain professional manner that will enhance the nursing students’ learning motivation and attempts to strive and encourage positive behaviors [8]. In addition, the family environment will affect nursing students’ concepts, behaviors, attitudes, and even personal future development. Part-time work experience will improve their self-esteem and psychological and physical adjustment, which enhances students’ knowledge of future work and helps them become more independent. Nursing students’ clinical work performance, independent character, interpersonal interaction, and self-confidence may be affected by their part-time work experience [9].

Adults can have intrinsic motivation for self-directed learning in any topic. They bring life experience and knowledge into their learning according to different roles and positions, attaching importance to practical operations and to goal-oriented and appropriate learning, which is the only way through which learning can be truly achieved. Adult learners like to be respected and, as such, the courses should be arranged, if possible, in relation to their life experiences. They can learn new knowledge, attitudes, and skills through dialog and interaction with teachers [10]. The design of clinical nursing education should be learner-centered, where the learner’s needs and limitations are understood, arranging and planning for pediatric learners is undertaken, and the concept of “therapeutic games” of nursing intervention can be integrated into teaching using vivid and distinctive patterns, health checks, treatments, health education, and other such teaching elements. In addition, nursing students should be taught to use therapeutic games in practical scenarios in order to enhance their interest and experience in childcare.

Hospitalization always puts great pressure on sick children. For sick children, hospitalization means that they will feel physically uncomfortable and restricted as they need to stay in an unfamiliar environment and undergo unfamiliar medical treatments, such as intravenous injections and the measurement of vital signs, which causes sick children to temporarily lose their sense of self-control [11]. According to the Ministry of Health and Welfare’s 2017 announcement, pneumonia ranked 7th among the top 10 causes of death in children and adolescents [12]. According to the 2020 announcement from the Directorate-General of Budget, Accounting and Statistics, Executive Yuan, respiratory diseases and gastrointestinal diseases ranked as the highest and the second highest among the top 10 diseases in children in 2018, respectively [13]. Nebulization therapy is a common treatment in pediatric care [14], and abdominal ultrasound is a key diagnostic support for unexplained acute abdomens [15,16] which also prevents the occurrence of medical disputes [17]. Less than 5% of pediatric abdominal emergencies are caused by diseases. When ruling out pathological factors, constipation caused by lifestyle is commonly seen. A small-volume enema is commonly used to help relieve constipation, which is a common examination and therapeutic activity in pediatrics [18].

Therapeutic games refer to developing a trust relationship through games, and it helps sick children express their feelings and deal with stressful moods, providing medical staff with the possibility of assessing sick children’s cognition and needs in medical scenarios, as well as assisting in adjustment [19]. They assist medical staff in observing sick children’s physical and mental needs and feelings, as well as helping sick children to express their fears and stress, improve their sense of security in order to face the treatment process, and cooperate with examinations and treatments [3]. Picture-based teaching books have been used in various pediatric clinical nursing practices: they can help to alleviate sick children’s discomfort during treatments, and for preschool children who are in the phase of picture-based cognitive development, they can be a therapeutic story and come with a theme on how to become a good guided medium for education [20]. In the systematic analysis of 16 studies by Brondani, J.P. and Pedro, E.N.R., it was found that the use of therapeutic games—which are explained through story guidance—in nursing intervention enables sick children and their family members to understand the nursing process, as well as helps to cure discomforts in the examination and treatment processes. This proves that it is a highly accepted, applicable communication method, which also has low cost, for children in clinic nursing [21].

The purpose of this study is to prove that AI-generated image teaching materials are more helpful in clinical communication with sick children and can also be used as effective teaching materials and tools for pediatric nursing students to improve their caring skills.

## 2. Materials and Methods

### 2.1. Study Design and Participants

This study is a quasi-experimental study that used AI image health education e-books created by the nursing teacher and students as therapeutic game teaching tools for the experimental group, thereby teaching the nursing students of the experimental group to intervene when sick children produce behavioral responses to fear. The nursing students of the control group were provided traditional paper-handout teaching materials to explain the theory and methods when using therapeutic games to intervene when sick children produce behavioral responses to fear. The nursing teachers of the experimental group and the control group conducted evaluations of the nursing students’ self-efficacy in dealing with sick children’s fear of examinations and treatments, as well as sick children’s behavioral responses to fear of examinations and treatments before and after intervention.

The research sample estimation method used the G-Power version 3.1 software for calculation and ANOVA with an effect size of 0.25 (medium), alpha level of 0.05, and power of 0.8. In the estimated sample collection, there were 34 subjects in each of the experimental and control groups, increasing admissions by 5% to a total of 72 subjects. The basic information, such as age, sex, level of education, related working experience, etc., of the two groups did not present significant differences when examined using an independent *t*-test and a chi-square test at a significance level of *p* > 0.05; thus, the two groups had similar conditions prior to intervention.

### 2.2. Measurement

#### 2.2.1. AI Image Health Education E-Books

Canva AI can transform complex health concepts into easy-to-understand charts and illustrations. Such visual presentations can help children grasp important information faster. At the same time, Canva AI can create interactive games and quizzes. These activities can not only improve children’s sense of participation but can also allow them to learn health knowledge through games, enhancing the learning effects. In addition, Canva AI can create stories and case sharing. This content can help children understand the importance of healthy behaviors and practice healthy behaviors. In summary, Canva AI significantly improves children’s understanding of health knowledge through the creation of visual teaching materials, interactive learning, and personalized content, and helps nurses deliver health information more effectively.

The nursing teacher applied Canva AI to generate AI image health education e-books on five topics of common examinations and treatments for children—namely, “vital signs measurement of children”, “bedside ultrasound of children”, “aerosol therapy of children”, “intravenous injection of children”, and “small volume enema of children (Figure 1). These were integrated with the learning needs of the nursing students, as well as in the examinations and treatments commonly seen in pediatric units and in literature reviews. The accuracy and suitability of the content and information conveyed in the AI image health education e-books were tested in terms of their validity by six nursing teachers with more than three years of clinical experience (CVI value of 1.0).

#### 2.2.2. Nursing Students’ Self-Efficacy Scale for Preschool Sick Children’s Fears of Medical Examinations and Treatments

This part was conducted using a revised version of the “Nursing Students’ Self-Efficacy Scale for Preschool Sick Children’s Fear of Medical Examination and Treatment”, which refers to the Chinese version of the General Self-Efficacy Scale [22]. The scale was rated by six nursing teachers with more than three years of clinical experience for content suitability and text clarity. Where CVI ≥ 0.8 was retained, 10 questions had a CVI value > 0.83–1, and the overall CVI value was 0.92. Using Cronbach’s α as a measure of the internal consistency reliability of the scale and a total of 10 questions, a pretest was conducted on 30 nursing students on the internship program, and the results showed that Cronbach’s α of the total scale was 0.809. The test was scored on a 4-point Likert scale, with 1 to 4, respectively, representing “not at all correct”, “somewhat correct”, “mostly correct”, and “completely correct”. The total score scale ranged from 0 to 40, and higher scores indicated greater self-efficacy.

#### 2.2.3. Observation Scale for Sick Children’s Behavioral Response to Fear of Medical Examinations and Treatments

The content’s validity was assessed by experts based on a literature review and clinical experience. Six nursing teachers with more than three years of clinical experience rated the content in terms of suitability and text clarity. A CVI ≥ 0.8 was retained, and 8 of the questions had a CVI value ≥ 0.8–1. The overall CVI value was 0.9. Six sick children undertook a test using the Observation Scale for Sick Children’s Behavioral Response to Fear of Medical Examination and Treatment, and the consistency among the observers’ observations on the behavioral responses to fear reached 100%. The internal consistency and reliability of the scale was tested with Cronbach’s α, and the 30 sick children’s behavior was observed. Cronbach’s α was 0.76, with a total of 8 scoring items. Each of the remaining items were worth 1 point, with a maximum score of 8 points. Higher scores indicate greater fear.

### 2.3. Research Time and Place

The study period was from 1 June 2022 to 31 July 2023. The subjects of this study were pediatric nursing students on an internship in a teaching hospital.

### 2.4. Ethical Considerations

This research project was reviewed and approved by the IRB of the China Medical University Hospital (no: CRREC-111-051). The research subjects could quit at any time and had the right to raise questions, and the questionnaires were anonymous. The questionnaire results were numbered anonymously in order to delink and ensure confidentiality. The names and conditions of the research subjects will never be publicized, and the results are for academic use only.

### 2.5. Data Collection and Analysis

After the data were collected, decoded, logged, and archived, statistical analysis was performed with the SPSS Version 26.0 statistical software package for Windows/PC. Statistical analysis was conducted based on the research purpose, and α = 0.05 was set as the standard for significant differences.

#### 2.5.1. One-Way Analysis of Covariance (ANCOVA)

It was required that the homogeneity of the within-group regression coefficient be tested in order to ensure that there were no interactions between the groups in the pretest before carrying out the one-way analysis of covariance (ANCOVA). This was required as differences may have existed between the control groups at the pretest stage.

#### 2.5.2. Generalized Estimating Equation (GEE)

When the data pertained to longitudinal data, the status of the same subject at different time points was recorded, and there was also correlation between the observation values from the same subject. As the same subjects in this study were observed at multiple time points (before and after intervention), they can be considered interdependent; therefore, this study is suitable for GEE analysis. As the dependent variable in this study is a numerical variable, it is statistically significant (*p* < 0.05) when the assumed distribution in the GEE is a normal distribution, the link function has identity, and the 95% CI of its β does not contain 0.

## 3. Results

### 3.1. Comparison Between Self-Efficacy of the Experimental Group and the Self-Efficacy of the Control Group After Intervention (ANCOVA)

As can be seen from Table 1, the test results of the homogeneity of the within-group regression coefficient for self-efficacy in the group × pretest did not reach the level of a statistically significant difference (F = 1.743, *p* > 0.05), which indicates that the assumption of the homogeneity of regression coefficients was not violated. Thus, a one-way analysis of covariance (ANCOVA) could be carried out directly.

As can be seen from Table 2, under the self-efficacy score before the control of interventions, the self-efficacy scores of the groups (experimental group vs. control group) reached a statistically significant difference (*p* < 0.05), which means that the self-efficacy score of the experimental group was higher than that of the control group (where the β coefficient was 0.356).

### 3.2. Comparison Between Experimental Group and Control Group on Behavioral Response to Fear After Intervention (ANCOVA)

As can be seen from Table 3, the test results of the homogeneity of the within-group regression coefficient for fear in the group × pretest did not reach the level of a statistically significant difference (F = 1.264, *p* > 0.05), which indicates that the assumption of the homogeneity of regression coefficients was not violated. Thus, one-way analysis of covariance (ANCOVA) could be carried out directly.

As can be seen from Table 4, regarding the fear before the control of the interventions, the fear scores of the groups (experimental group vs. control group) reached the level of a statistically significant difference (*p* < 0.05), which means that the fear score of the experimental group was lower than that of the control group (where the β coefficient was −1.540).

### 3.3. Comparison of the Difference Between the Experimental Group’s Impact of Self-Efficacy on the Behavioral Responses to Fear and the Control Group’s Impact of Self-Efficacy on the Behavioral Responses to Fear (GEE)

Next, generalized estimating equations (GEEs) were used to conduct an analysis. First, the aim was to understand whether the self-efficacy of the experimental group and the control group had a significant impact on fear; second, the aim was to further compare whether there was a significant difference in the impact of self-efficacy on the fear between the two groups.

As can be seen from Table 5, the experimental group’s self-efficacy had a statistically significant impact on fear (*p* < 0.05), and the higher the experimental group’s self-efficacy, the lower the fear (where the β coefficient was −0.914); in addition, the control group’s self-efficacy had no statistically significant impact on fear (*p* > 0.05).

## 4. Discussion

The findings of this study were deemed to be consistent with the studies of Huang et al.; Chang et al.; Natalya A. et al.; Holliman, R.P. and Blanco, P.J.; Bliss, H. and Gildner, J., who showed that therapeutic games can significantly improve sick children’s behavioral responses to fear [23,24,25,26,27]. AI image health education e-books should be used to teach nursing students how to use therapeutic games. One-way analysis of covariance (ANCOVA) was used to find that the self-efficacy scores of the experimental group in dealing with the sick children’s fear of examinations and treatments were higher when compared with that of the control group (where the β coefficient was 0.356), and the difference was also found to be statistically significant (*p* < 0.05). The score of the sick children’s behavioral responses to fear in the experimental group was lower than that in the control group (where the β coefficient was −1.540), which reached the level of a statistically significant difference (*p* < 0.05). The self-efficacy of the experimental group in dealing with the sick children’s fear of examinations and treatments had a statistically significant impact on the sick children’s behavioral responses to fear of examinations and treatments (*p* < 0.05), and the higher self-efficacy of the experimental group in dealing with sick children’s fear of examinations and treatments led to lower sick children’s behavioral responses to fear of examinations and treatments (where the β coefficient was −0.914). The self-efficacy of the control group in dealing with the sick children’s fear of examinations and treatments did not have a statistically significant impact on the sick children’s behavioral response to fear of examinations and treatments (*p >* 0.05); therefore, this study also demonstrates the benefits of applying AI-generated image teaching materials in clinical education [28].

## 5. Conclusions

This study proved that AI image health education e-books have a better effect than traditional paper handout teaching materials in improving nursing students’ self-efficacy in using therapeutic games to deal with sick children’s fear of examinations and treatments and in reducing sick children’s behavioral responses to fear of examinations and treatments. AI-driven tools can enable the development of personalized e-learning materials that target specific areas for cognitive improvement. This targeted approach can enhance knowledge retention and skill development, resulting in better-prepared healthcare professionals.

Due to the limitations of the research discipline and the samples in this study, which affect the inference of this study’s results, it is recommended to carry out further studies on different disciplines and samples to explore the effectiveness of AI applications in medical and nursing-related education [28,29].

## Figures and Tables

**Figure 1 bioengineering-11-01097-f001:**
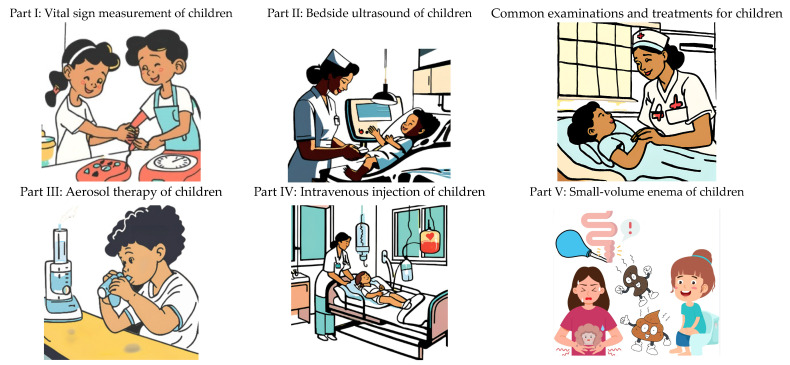
Five topics of AI image health education e-books on common examinations and treatments for children.

**Table 1 bioengineering-11-01097-t001:** Test of the homogeneity of the within-group regression coefficient for self-efficacy.

Source of Variation	Sum of Squares (*SS*)	Degrees of Freedom (*df*)	Mean Square (*MS*)	*F*	*p-Value*
Group	0.691	1	0.691	4.514	0.037
Pretest	9.992	1	9.992	65.303	<0.001
Group × Pretest	0.267	1	0.267	1.743	0.191
Error	11.017	72	0.153		

**Table 2 bioengineering-11-01097-t002:** Comparison of the self-efficacy of the experimental group and the self-efficacy of the control group after intervention.

Variable Name	*β*	*SE*	*p-Value*	95% *C.I.*	
				*Lower*	*Upper*
Self-Efficacy before Intervention	0.805	0.097	<0.001	0.612	0.999
Group (Experimental Group vs. Control Group)	0.356	0.090	<0.001	0.176	0.537

Dependent Variable: Self-Efficacy after Intervention.

**Table 3 bioengineering-11-01097-t003:** Test of the homogeneity of the within-group regression coefficient for fear.

Source of Variation	Sum of Squares (*SS*)	Degrees of Freedom (*df*)	Mean Square (*MS*)	*F*	*p-Value*
Group	0.102	1	0.102	0.040	0.842
Pretest	29.356	1	29.356	11.486	0.001
Group × Pretest	3.231	1	3.231	1.264	0.265
Error	184.021	72	2.556		

**Table 4 bioengineering-11-01097-t004:** Comparison of the fear of the experimental group and the fear of the control group after intervention.

Variable Name	*β*	*SE*	*p-Value*	95% *C.I.*	
				*Lower*	*Upper*
Fear before Intervention	0.404	0.125	0.002	0.154	0.654
Group (Experimental Group vs. Control Group)	−1.540	0.374	<0.001	−2.285	−0.794

Dependent Variable: Fear after Intervention.

**Table 5 bioengineering-11-01097-t005:** The experimental group’s and the control group’s impact of self-efficacy on fear.

Variable Name	*β*	*SE*	*p-Value*	95% *C.I.*	
				*Lower*	*Upper*
Time Points					
After Intervention	3.291	0.883	<0.001	1.560	5.021
Before Intervention	5.356	0.782	<0.001	3.824	6.888
Group (Experimental Group vs. Control Group)	0.667	1.104	0.546	−1.497	2.831
Group × Self-Efficacy					
Experimental Group × Self-Efficacy	−0.914	0.351	0.009	−1.602	−0.226
Control Group × Self-Efficacy	−0.224	0.342	0.512	−0.896	0.447

Dependent Variable: Fear.

## Data Availability

All of the relevant datasets in this study are described in the manuscript.

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
