# Peer review of "The Effectiveness of Applying Artificial Intelligence in Sick Children’s Communication"

_bioengineering, 2024, doi:10.3390/bioengineering11111097_

Round 1
Reviewer 1 Report
Comments and Suggestions for Authors
The manuscript can be improved in following ways:
1. The abstract touches on various aspects of AI in communication, but the primary research goal is not clearly stated. Refine it to give a precise objective upfront, highlighting what the study aims to achieve.
2. The Introduction section uses complex phrasing that could be simplified to enhance readability. For example, sentences like "exhibiting behavioral responses to fears, such as crying and rejection" could be shortened to "exhibiting fearful behaviors such as crying and rejection."
3. In the Materials and Methods section, more details should be provided regarding why the control group was chosen and how the handouts provided were comparable to AI tools. This helps in understanding the rationale for comparison.
4. It is unclear why these specific AI e-books were chosen for the study. Provide some background on how they were developed and why they are considered effective for the targeted cognitive improvement.
5. The explanation of the self-efficacy scale is a bit technical. Simplify it or add a footnote explaining how to interpret the Cronbach’s alpha values and why they are important for reliability.
6. The Generalized Estimating Equation (GEE) method is mentioned but not thoroughly explained. Offering a brief description of why it was used and how it applies to the study’s data would enhance clarity for readers unfamiliar with this method.
7. The figure showing the AI picture-based teaching e-books is too simplistic. A more detailed caption explaining the relevance of each e-book to pediatric care and the study's aims would be helpful.
8. There is no mention of how potential bias was controlled during data collection. A brief discussion on how the anonymity of participants was maintained and how researcher bias was minimized would strengthen the methodology.
9.While the study is well-structured, the limitations are not discussed in enough detail. For instance, the small sample size and the generalizability of results to other settings or populations should be addressed.
10. Several sentences throughout the manuscript, such as "common examinations and treatments for children of five AI picture-based teaching e-books," are awkwardly phrased. A thorough proofreading and grammar check is necessary to smooth out readability.
11. Please cite the following relevant reviews if you want:
(a) DOI: 10.1109/ACCESS.2024.3359906
(b) DOI: 10.1038/s41378-023-00555-7
Author Response
The Effectiveness of Applying Artificial Intelligence on Sick Children’s Communication
Response to Reviewer A Comments
Comments 1,
The abstract touches on various aspects of AI on communication, but the primary research goal is not clearly stated. Refine it to give a precise objective upfront, highlighting what the study aims to achieve.
Response 1,
The paragraphs in the red field in the abstract describe the results of this research and prove the purpose of this research:
This quasi-experimental study proves that the application of AI image health education e-books by the nursing teachers are more effective than narrative handouts in improving nursing students’ self-efficacy when using therapeutic games to deal with and reduce sick children’s fears of medical examinations and treatments (p < 0.05).
Comments 2,
The Introduction section uses complex phrasing that could be simplified to enhance readability. For example, sentences like "exhibiting behavioral
responses to fears, such as crying and rejection" could be shortened to "exhibiting fearful behaviors such as crying and rejection."
Response 2,
Corrections in red text in the foreword of the manuscript:
The majority of pediatric care patients are in their early-childhood and preschool years; thus, they are more prone to exhibiting fearful behaviors such as crying and rejection,……………………………
Comments or questions 3,
In the Materials and Methods section, more details should be provided regarding why the control group was chosen and how the handouts provided were comparable to AI tools. This helps in understanding the rationale for comparison.
Response 3,
Compare the experimental group’s use of technology to produce AI teaching materials, and the control group’s continued use of traditional paper handout teaching materials on children’s understanding and effectiveness in reducing anxiety and fear:
- Materials and Methods
2.1. Study Design and Participants
This study is a quasi-experimental study that used AI image health education e-books created by the nursing teachers and students as therapeutic game teaching tools for the experimental group, thereby teaching the nursing students of the experimental group to intervene when sick children produce behavioral responses to fear. The nursing students of the of the control group provided traditional paper handout teaching materials to explain the theory and methods when using therapeutic games that might be used to intervene when sick children produce behavioral responses to fear. The nursing teachers of the experimental group and the control group conducted evaluations of nursing students’ self-efficacy in dealing with sick children’s fear of examinations and treatments, as well as sick children’s behavioral responses to fear of examinations and treatments before and after intervention.
Comments or questions 4,
It is unclear why these specific AI e-books were chosen for the study. Provide some background on how they were developed and why they are considered effective for the targeted cognitive improvement.
Response 4,
- The basis for the development of AI image health education e-books and the establishment of expert validity are as follows (red field colony):
2.2.1. AI image health education e-books
Five topics of AI image health education e-books on common examinations and treatments for children—namely, “vital signs measurement of children,” “bedside ultrasound of children,” “aerosol therapy of children,” “intravenous injection of children,” and “small volume enema of children”—were used. These were integrated with the learning needs of nursing students, as well as in the examinations and treatments commonly seen in pediatric units and in literature reviews. The accuracy and suitability of the content and information conveyed in the AI image health education e-books were tested in terms of their validity by six nursing teachers with more than three years of clinical experience (CVI value of 1.0).
- Please see the 3C GA of The Effectiveness of Applying Artificial Intelligence on Sick Children’s Communication.
Comments or questions 5,
The explanation of the self-efficacy scale is a bit technical. Simplify it or add a footnote explaining how to interpret the Cronbach’s alpha values and why they are important for reliability.
Response 5,
Reliability refers to the stability and consistency of the measurement tool, the higher the reliability, the better the degree to which the research tool can be trusted. Cronbach’s alpha coefficient is a very common reliability method for detecting attitude and behavior scales, the greater the Cronbach’s alpha coefficient, indicating the better the internal consistency of the scale.
Comments 6,
The Generalized Estimating Equation (GEE) method is mentioned but not thoroughly explained. Offering a brief description of why it was used and how it applies to the study’s data would enhance clarity for readers unfamiliar with this method.
Response 6,
The Generalized linear model (Generalized Estimating Equation, GEE): Since the same research subject in this study observed multiple time points (before intervention, after intervention), there is dependence, so this study is suitable for GEE analysis. Since the dependent variable is a numerical variable, it is statistically significant (p<0.05) when the assumed distribution in GEE is normal distribution and the link function is identity, and the 95% CI of its β does not include 0.
Comments 7,
The figure showing the AI picture-based teaching e-books is too simplistic. A more detailed caption explaining the relevance of each e-book to pediatric care and the study's aims would be helpful.
Response 7,
Figure 1 has been corrected as suggested, please see pages 4 of the manuscript.
Comments or questions 8,
There is no mention of how potential bias was controlled during data collection. A brief discussion on how the anonymity of participants was maintained and how researcher bias was minimized would strengthen the methodology.
Response 8,
Since there is only one researcher and there is no inter-researcher difference, the potential differences in the pretest data between the experimental group and the control group are controlled and the data are analyzed using the one-way analysis of covariance (ANCOVA) statistical method.
Comments 9,
While the study is well-structured, the limitations are not discussed in enough detail. For instance, the small sample size and the generalizability of results to other settings or populations should be addressed.
Response 9,
The conclusions in the manuscript add limitations of this study and suggestions for further research:
- Conclusions
This study proved that AI image health education e-books have a better effect than traditional paper handout teaching materials in improving nursing students’ self-efficacy in using therapeutic games to deal with sick children’s fear of examinations and treatments and in reducing sick children’s behavioral responses to fear of examinations and treatments. AI-driven tools can enable the development of customized e-learning materials that target specific areas for cognitive improvement. This targeted approach can enhance knowledge retention and skill development, resulting in better prepared healthcare professionals.
Due to the limitations of research samples and discipline in this study, which affects the inference of this study, it is recommended to increase further research on different disciplines and samples to explore the effectiveness of AI applications in different aspects of medical and nursing education.
Comments 10,
Several sentences throughout the manuscript, such as "common examinations and treatments for children of five AI picture-based teaching e-books," are awkwardly phrased. A thorough proofreading and grammar check is necessary to smooth out readability.
Response 10,
"common examinations and treatments for children of five AI image health education e-books," in the revised manuscript are “Five topics of AI image health education e-books on common examinations and treatments for children.”
Comments 11,
Please cite the following relevant reviews if you want:
(a) DOI: 10.1109/ACCESS.2024.3359906
(b) DOI: 10.1038/s41378-023-00555-7
Response 11,
Document numbers 15. DOI: 10.1109/ACCESS.2024.3359906 and 16. DOI: 10.1038/s41378-023-00555-
have been added to the literature.
- Moisello, E., Novaresi, L., Sarkar, E., Malcovati, P., Costa, T. L., & Bonizzoni, E. PMUT and CMUT Devices for Biomedical Applications: A Review. IEEEAccess. 2023, 12, 18640-18657. DOI: 10.1109/ACCESS.2024.3359906
- Roy, K., Lee, J. E., & Lee, C. Thin-film PMUTs: a review of over 40 years of research. Microsystems & Nanoengineering. 2023, 9(95),1-17. DOI: 10.1038/s41378-023-00555-7
Thank you very much.

Reviewer 2 Report
Comments and Suggestions for Authors
The paper is interesting and has a potential for publication provided that the following points are all taken seriously and responded accordingly:
- it is not clear to me if the contents of this article fit with the topic of the special issue: communication with sick children seems different from "Advanced Measurement in Biomedical Engineering: Integration Motion Tracking, Virtual Reality, Artificial Intelligence, and Biosensors for Sports Healthcare" . Please motivate somewhere in the Introduction.
- the use of AI here is somewhat minimal. I understand that picture to be shown to children are AI-generated. Nonetheless I would like to understand more: with which AI tool? Based on which selection criterion? Etc Please explain better.
- Again on the AI perspective, much more could be done in this field. For example, researchers could use automatic recognition of motion tracking from recorded videos to try to interpret children' reactions to pictures or other; moreover their human experts' efforts in the interpretation could be augmented with the support of specific AI tools. As to motion tracking and automatic video interpretation, there is a lot of specializing literature.I would suggest the author for example to cite the following paper (even if in another application field), as well as to discuss this general point in the Conclusion, something that sounds like this, we use a minimal AI so far but more can be done ... paper: AA VV,First responders' crystal ball: How to scry the emergency from a remote vehicle, Proc of 27th IEEE International Performance Computing and Communications Conference, 556-561, doi: 10.1109/PCCC.2007.358940
- Finally I would also expect an attempt to better explain and motivate the hypothesis testing procedures conducted and reported in the Results Section. An average reader of this SI could get confused in absence of clearer explanations
If all these point are fixed the paper could warrant publication.
Author Response
The Effectiveness of Applying Artificial Intelligence on Sick Children’s Communication
Response to Reviewer B Comments
The paper is interesting and has a potential for publication provided that the following points are all taken seriously and responded accordingly:
Comments 1,
- it is not clear to me if the contents of this article fit with the topic of the special issue: communication with sick children seems different from "Advanced Measurement in Biomedical Engineering: Integration Motion Tracking, Virtual Reality, Artificial Intelligence, and Biosensors for Sports Healthcare" . Please motivate somewhere in the Introduction.
Response 1,
1.This study demonstrates that images can help understand language, text and health education.2.The purpose of this study is to prove that artificial intelligence-generated image teaching materials are more helpful in clinical communication with sick children, and can also be used as effective teaching materials and tools for pediatric nursing students to improve their caring skills.
Comments 2,
- the use of AI here is somewhat minimal. I understand that picture to be shown to children are AI-generated. Nonetheless I would like to understand more: with which AI tool? Based on which selection criterion? Etc Please explain better.
- Again on the AI perspective, much more could be done in this field. For example, researchers could use automatic recognition of motion tracking from recorded videos to try to interpret children' reactions to pictures or other; moreover their human experts' efforts in the interpretation could be augmented with the support of specific AI tools. As to motion tracking and automatic video interpretation, there is a lot of specializing literature. I would suggest the author for example to cite the following paper (even if in another application field), as well as to discuss this general point in the Conclusion, something that sounds like this, we use a minimal AI so far but more can be done ... paper: AA VV, First responders' crystal ball: How to scry the emergency from a remote vehicle, Proc of 27th IEEE International Performance Computing and Communications Conference, 556-561, doi: 10.1109/PCCC.2007.358940
- Finally I would also expect an attempt to better explain and motivate the hypothesis testing procedures conducted and reported in the Results Section. An average reader of this SI could get confused in absence of clearer explanations
If all these point are fixed the paper could warrant publication.
Response 2,The preface and research methods in the corrected manuscript have emphasized the research purpose and research tools.
Thank you very much.

Round 2
Reviewer 1 Report
Comments and Suggestions for Authors
I find the changes correctly implemented. I think it is good to go.
Author Response
Reviewer’s comment 1,
The abstract touches on various aspects of AI on communication, but the primary research goal is not clearly stated. Refine it to give a precise objective upfront, highlighting what the study aims to achieve.
Author's corresponding response 1,
The paragraphs in the red field in the abstract describe the results of this research and prove the purpose of this research:
This quasi-experimental study proves that the application of AI image health education e-books by the nursing teachers are more effective than narrative handouts in improving nursing students’ self-efficacy when using therapeutic games to deal with and reduce sick children’s fears of medical examinations and treatments (p < 0.05).
Reviewer’s comment 2,
The Introduction section uses complex phrasing that could be simplified to enhance readability. For example, sentences like "exhibiting behavioral
responses to fears, such as crying and rejection" could be shortened to "exhibiting fearful behaviors such as crying and rejection."
Author's corresponding response 2,
Corrections in red text in the foreword of the manuscript:
The majority of pediatric care patients are in their early-childhood and preschool years; thus, they are more prone to exhibiting fearful behaviors such as crying and rejection,……………………………
Reviewer’s comment 3,
In the Materials and Methods section, more details should be provided regarding why the control group was chosen and how the handouts provided were comparable to AI tools. This helps in understanding the rationale for comparison.
Author's corresponding response 3,
Compare the experimental group’s use of technology to produce AI teaching materials, and the control group’s continued use of traditional paper handout teaching materials on children’s understanding and effectiveness in reducing anxiety and fear:
- Materials and Methods
2.1. Study Design and Participants
This study is a quasi-experimental study that used AI image health education e-books created by the nursing teachers and students as therapeutic game teaching tools for the experimental group, thereby teaching the nursing students of the experimental group to intervene when sick children produce behavioral responses to fear. The nursing students of the of the control group provided traditional paper handout teaching materials to explain the theory and methods when using therapeutic games that might be used to intervene when sick children produce behavioral responses to fear. The nursing teachers of the experimental group and the control group conducted evaluations of nursing students’ self-efficacy in dealing with sick children’s fear of examinations and treatments, as well as sick children’s behavioral responses to fear of examinations and treatments before and after intervention.
Reviewer’s comment 4,
It is unclear why these specific AI e-books were chosen for the study. Provide some background on how they were developed and why they are considered effective for the targeted cognitive improvement.
Author's corresponding response 4,
- The basis for the development of AI image health education e-books and the establishment of expert validity are as follows (red field colony):
2.2.1. AI image health education e-books
Five topics of AI image health education e-books on common examinations and treatments for children—namely, “vital signs measurement of children,” “bedside ultrasound of children,” “aerosol therapy of children,” “intravenous injection of children,” and “small volume enema of children”—were used. These were integrated with the learning needs of nursing students, as well as in the examinations and treatments commonly seen in pediatric units and in literature reviews. The accuracy and suitability of the content and information conveyed in the AI image health education e-books were tested in terms of their validity by six nursing teachers with more than three years of clinical experience (CVI value of 1.0).
- Please see the Graphical Abstract.
Reviewer’s comment 5,
The explanation of the self-efficacy scale is a bit technical. Simplify it or add a footnote explaining how to interpret the Cronbach’s alpha values and why they are important for reliability.
Author's corresponding response 5,
Reliability refers to the stability and consistency of the measurement tool, the higher the reliability, the better the degree to which the research tool can be trusted. Cronbach’s alpha coefficient is a very common reliability method for detecting attitude and behavior scales, the greater the Cronbach’s alpha coefficient, indicating the better the internal consistency of the scale.
Reviewer’s comment 6,
The Generalized Estimating Equation (GEE) method is mentioned but not thoroughly explained. Offering a brief description of why it was used and how it applies to the study’s data would enhance clarity for readers unfamiliar with this method.
Author's corresponding response 6,
The Generalized linear model (Generalized Estimating Equation, GEE): Since the same research subject in this study observed multiple time points (before intervention, after intervention), there is dependence, so this study is suitable for GEE analysis. Since the dependent variable is a numerical variable, it is statistically significant (p<0.05) when the assumed distribution in GEE is normal distribution and the link function is identity, and the 95% CI of its β does not include 0.
Reviewer’s comment 7,
The figure showing the AI picture-based teaching e-books is too simplistic. A more detailed caption explaining the relevance of each e-book to pediatric care and the study's aims would be helpful.
Author's corresponding response 7,
Figure 1 has been corrected as suggested, please see pages 4 of the manuscript.
Reviewer’s comment 8,
There is no mention of how potential bias was controlled during data collection. A brief discussion on how the anonymity of participants was maintained and how researcher bias was minimized would strengthen the methodology.
Author's corresponding response 8,
Since there is only one researcher and there is no inter-researcher difference, the potential differences in the pretest data between the experimental group and the control group are controlled and the data are analyzed using the one-way analysis of covariance (ANCOVA) statistical method.
Reviewer’s comment 9,
While the study is well-structured, the limitations are not discussed in enough detail. For instance, the small sample size and the generalizability of results to other settings or populations should be addressed.
Author's corresponding response 9,
The conclusions in the manuscript add limitations of this study and suggestions for further research:
- Conclusions
This study proved that AI image health education e-books have a better effect than traditional paper handout teaching materials in improving nursing students’ self-efficacy in using therapeutic games to deal with sick children’s fear of examinations and treatments and in reducing sick children’s behavioral responses to fear of examinations and treatments. AI-driven tools can enable the development of customized e-learning materials that target specific areas for cognitive improvement. This targeted approach can enhance knowledge retention and skill development, resulting in better prepared healthcare professionals.
Due to the limitations of research discipline and samples in this study, which affects the inference of this study results, it is recommended to increase further studies on different disciplines and samples to explore the effectiveness of AI applications in medical and nursing related education [28, 29].
Reviewer’s comment 10,
Several sentences throughout the manuscript, such as "common examinations and treatments for children of five AI picture-based teaching e-books," are awkwardly phrased. A thorough proofreading and grammar check is necessary to smooth out readability.
Author's corresponding response 10,
"common examinations and treatments for children of five AI image health education e-books," in the revised manuscript are “Five topics of AI image health education e-books on common examinations and treatments for children.”
Reviewer’s comment 11,
Please cite the following relevant reviews if you want:
(a) DOI: 10.1109/ACCESS.2024.3359906
(b) DOI: 10.1038/s41378-023-00555-7
Author's corresponding response 11,
Document numbers 15. DOI: 10.1109/ACCESS.2024.3359906 and 16. DOI: 10.1038/s41378-023-00555-
- have been added to the literature.
- Moisello, E.; Novaresi, L.; Sarkar, E.; Malcovati, P.; Costa, T. L.; Bonizzoni, E. PMUT and CMUT Devices for Biomedical
Applications: A Review. IEEEAccess. 2023, 12, 18640-18657. DOI: 10.1109/ACCESS.2024.3359906
- Roy, K.; Lee, J. En-Yuan; Lee, C. Thin-film PMUTs: a review of over 40 years of research. Microsystems & Nanoengineering. 2023, 9(95),1-17. DOI: 10.1038/s41378-023-00555-7
Thank you very much.

Reviewer 2 Report
Comments and Suggestions for Authors
I am NOT satisfied at all by this revision. The reason is simple and it can be simply understood by comparing the comments I have done in my first review (see below) and the response by the authors (see at the bottom). If authors do not have any intention to carefully take care of my comments my role is not useful.
Comments 2, (REVIEWER)
- the use of AI here is somewhat minimal. I understand that picture to be shown to children are AI-generated. Nonetheless I would like to understand more: with which AI tool? Based on which selection criterion? Etc Please explain better.
- Again on the AI perspective, much more could be done in this field. For example, researchers could use automatic recognition of motion tracking from recorded videos to try to interpret children' reactions to pictures or other; moreover their human experts' efforts in the interpretation could be augmented with the support of specific AI tools. As to motion tracking and automatic video interpretation, there is a lot of specializing literature. I would suggest the author for example to cite the following paper (even if in another application field), as well as to discuss this general point in the Conclusion, something that sounds like this, we use a minimal AI so far but more can be done ... paper: AA VV, First responders' crystal ball: How to scry the emergency from a remote vehicle, Proc of 27th IEEE International Performance Computing and Communications Conference, 556-561, doi: 10.1109/PCCC.2007.358940
- Finally I would also expect an attempt to better explain and motivate the hypothesis testing procedures conducted and reported in the Results Section. An average reader of this SI could get confused in absence of clearer explanations
If all these point are fixed the paper could warrant publication.
Response 2, (AUTHORS) The preface and research methods in the corrected manuscript have emphasized the research purpose and research tools.
Author Response
Reviewer’s comment 1,
- it is not clear to me if the contents of this article fit with the topic of the special issue: communication with sick children seems different from "Advanced Measurement in Biomedical Engineering: Integration Motion Tracking, Virtual Reality, Artificial Intelligence, and Biosensors for Sports Healthcare". Please motivate somewhere in the Introduction.
Author's corresponding response 1,
1. This study demonstrates that images can help understand language, text and health education.
2. The purpose of this study is to prove that artificial intelligence-generated image teaching materials are more helpful in clinical communication with sick children, and can also be used as effective teaching materials and tools for pediatric nursing students to improve their caring skills.
Reviewer’s comment 2,
- the use of AI here is somewhat minimal. I understand that picture to be shown to children are AI-generated. Nonetheless I would like to understand more: with which AI tool? Based on which selection criterion? Etc Please explain better.
- Again on the AI perspective, much more could be done in this field. For example, researchers could use automatic recognition of motion tracking from recorded videos to try to interpret children' reactions to pictures or other; moreover their human experts' efforts in the interpretation could be augmented with the support of specific AI tools. As to motion tracking and automatic video interpretation, there is a lot of specializing literature. I would suggest the author for example to cite the following paper (even if in another application field), as well as to discuss this general point in the Conclusion, something that sounds like this, we use a minimal AI so far but more can be done ... paper: AA VV, First responders' crystal ball: How to scry the emergency from a remote vehicle, Proc of 27th IEEE International Performance Computing and Communications Conference, 556-561, doi: 10.1109/PCCC.2007.358940
- Finally I would also expect an attempt to better explain and motivate the hypothesis testing procedures conducted and reported in the Results Section. An average reader of this SI could get confused in absence of clearer explanations
If all these point are fixed the paper could warrant publication.
Author's corresponding response 2,
1. This research method describes the application of Canva AI to generate pictures to create AI image health education e-books to communicate with pediatric patients about examinations and treatments. 2. Statistical analysis applications One-Way Analysis of Covariance (ANCOVA) and Generalized Estimating Equation (GEE) verify the effectiveness of using AI image health education e-books to communicate with children.
3.The conclusion of this study mentions the limitations of research discipline and samples in this study, which affects the inference of this study results. It is recommended to expand the field and sample number in future studies of medical and nursing related education to verify the effectiveness of AI applications.
4.Document numbers 29. doi: 10.1109/PCCC.2007.358940 have been added to the literature.
29. Roccetti, M.; Gerla, M; Palazzi, G.E.; Ferretti, S.; Pau, G. First Responders’ Crystal Ball: How to Scry the Emergency from a Remote Vehicle. Proc of 27th IEEE International Performance Computing and Communications Conference, 556-561. 2007. doi: 1109/PCCC.2007.358940
Thank you very much.
